# Fabrication of a Polybutylene Succinate (PBS)/Polybutylene Adipate-Co-Terephthalate (PBAT)-Based Hybrid System Reinforced with Lignin and Zinc Nanoparticles for Potential Biomedical Applications

**DOI:** 10.3390/polym14235065

**Published:** 2022-11-22

**Authors:** Asanda Mtibe, Lerato Hlekelele, Phumelele E. Kleyi, Sudhakar Muniyasamy, Nomvuyo E. Nomadolo, Osei Ofosu, Vincent Ojijo, Maya J. John

**Affiliations:** 1Centre for Nanostructures and Advanced Materials, DSI-CSIR Nanotechnology Innovation Centre, Council for Scientific and Industrial Research, Pretoria 0001, South Africa; 2Department of Chemistry, Faculty of Sciences, Nelson Mandela University, Port Elizabeth 6031, South Africa; 3Department of Textile Science, Faculty of Sciences, Nelson Mandela University, Port Elizabeth 6031, South Africa

**Keywords:** biopolymer blends, lignin, zinc nanoparticles, properties, biomedical applications

## Abstract

Polybutylene adipate-co-terephthalate (PBAT) was used in an effort to improve the properties of polybutylene succinate (PBS). The resultant blend consisting of PBS/PBAT (70/30) was reinforced with lignin at different loadings (5 to 15 wt.%) and zinc (ZnO) nanoparticles (1.5 wt.%). Hot melt extrusion and injection moulding were used to prepare the hybrid composites. The mechanical, thermal, physical, self-cleaning, and antimicrobial properties of the resultant hybrid composites were investigated. The transmission electron microscopy (TEM) results confirmed that ZnO was successfully prepared with average diameters of 80 nm. Fourier transform infrared (FTIR) spectroscopy and X-ray diffraction (XRD) confirmed that there were interactions between the fillers and the blend. The tensile strength and elongation at the break of the resultant materials decreased with increasing the loadings, while the tensile modulus showed the opposite trend. The melting behaviour of the blend was practically unaffected by incorporating lignin and ZnO nanoparticles. In addition, the incorporation of fillers reduced the thermal stability of the materials. Furthermore, the incorporation of ZnO nanoparticles introduced photocatalytic properties into the polymer blend, rendering it to be a functional self-cleaning material and enhancing its antimicrobial activities.

## 1. Introduction

There is a growing demand for the discovery and development of novel polymer-based materials for biomedical applications [1,2,3,4,5,6,7,8,9,10]. The polymer-based materials that are widely used for biomedical applications are derived from petroleum-based resources. Even though these materials have a high market value, their production has demonstrated a vast environmental pollution due to their non-biodegradable nature and they remain in the environment for the longest time. In addition, there is an anticipated depletion of petroleum-based resources in the near future [5,11,12,13]. In recent years, the scientific community have been focussing on the development of sustainable and eco-friendly alternatives to conventional polymer-based materials. In this respect, several biodegradable polymers (biopolymers) have been considered as a suitable alternative for conventional petroleum-based materials, and they have been extensively investigated for a range of applications [2,14,15,16]. This is due to their biocompatibility; sustainability; excellent mechanical properties which are comparable to those of conventional petroleum-based resources; biodegradability; non-toxicity; and non-immunogenicity [2,3].

Among the variety of biopolymers that have been extensively investigated over the past decades, polybutylene succinate (PBS) has attracted considerable interest due to its extraordinary properties such as biocompatibility, easy processability, biodegradability, non-toxicity, good mechanical and thermal properties, and its relatively high production capacity [3]. These unique properties render PBS as a potential material to replace petroleum-based resources in the near future for a wide range of potential applications. Unfortunately, PBS has shown some limitations such as poor ductility. Microbiological corrosion still exists in PBS, which limits their applications, as well as having a high production cost in comparison with conventional polymers [17].

Blending PBS with high ductile biopolymers is considered to be a promising strategy to address the stiffness of PBS without compromising its biodegradability feature. Among the high ductile biopolymers, PBAT is regarded as a suitable biopolymer for blending with PBS. Besides being very tough and its high ductility, other advantages of PBAT include easy processability, biodegradability, biocompatibility, and a high production capacity [18]. Numerous researchers have developed PBS and PBAT blends and have investigated their properties [19,20]. For instance, Costa et al. [17] fabricated PBS and PBAT blends and investigated their properties. The results showed that the properties of PBS can be tuned by blending with PBAT.

Although the blending of biopolymers is regarded as a suitable alternative, their high production cost limits their applications. To reduce the cost of biopolymers, the incorporation of inexpensive biodegradable fillers into biopolymers and the biopolymer blends is considered to be a promising approach to balance the performance and cost of the resultant materials. Among the biodegradable fillers, lignin has been a research hotspot recently. It is considered as the most abundantly available biopolymer on Earth, after cellulose [5]. However, lignin reinforced biopolymers are used to reduce the cost of biopolymers and to improve their performance [5,18,21,22,23,24,25,26,27,28,29]. Lignin has the potential to be used in biomedical applications. In addition, lignin is comprised of antimicrobial and antioxidant properties, which are important for potential biomedical applications wherein biocompatibility and biodegradability features are of great importance. For instance, Domínguez-Robles et al. [5] reported that the incorporation of lignin into PBS at various loadings was able to reduce the 2,2 diphenyl-1-picrylhydrazyl) (DPPH) concentrations up to 80% in less than 5 h, and demonstrated antimicrobial properties against *Staphylococcus aureus*. However, the incorporation of nanoparticles is necessary in order to extend their applications. Xiao et al. [27] highlighted in their study that the reinforcement of PBAT with both lignin and zinc nanoparticles enhanced the mechanical, antioxidant, UV shielding, and antimicrobial properties of the material.

To the best of our understanding, there are few studies that have reported on biopolymer reinforced lignin for biomedical applications. Self-cleaning and antimicrobial properties are crucial for biomedical applications. Therefore, inorganic fillers such as zinc nanoparticles have been used to retard the antimicrobial growth and potential for self-cleaning [13,30,31].

This study is aimed at fabricating a PBS/PBAT hybrid composite reinforced with lignin and ZnO for potential biomedical applications. To the best of our knowledge, the development of this hybrid composite has been fabricated here for the first time. Preliminary work on the optimisation of PBS/PBAT blends was conducted. The best performing blend was selected and incorporated with lignin and zinc nanoparticles. In this approach, different amounts of lignin and their blend were varied, while the zinc nanoparticles were constant. The effect of both lignin and zinc nanoparticles on the thermal, physical, mechanical, antimicrobial, and self-cleaning properties of the resultant hybrid composites were investigated.

## 2. Experimental Methods

### 2.1. Materials

Table 1 presents the biopolymers used in this work. SAPPI (KwaZulu Natal, South Africa) kindly provided lignosulphonate. Zinc acetate dihydrate, ethanol, polyethylene glycol, ammonium hydroxide, and formic acid were purchase from Sigma-Aldrich (Johannesburg, South Africa).

### 2.2. Synthesis of Zinc Oxide (ZnO) Nanoparticles

The method used for the synthesis of ZnO nanoparticles was adopted from our previous report, with some modifications [32]. Here, 80 g of zinc acetate dihydrate was dissolved in 1.5 L of ethanol by stirring magnetically until all of the solid was dissolved. To the clear Zn^2+^ solution, 150 g of polyethylene glycol was added as a capping agent and stabilizer. The mixture was then heated to 90 °C while stirring until all the polymer was dissolved, forming a melt. The Zn^2+^ ions were then precipitated by adding ammonium hydroxide to the mixture dropwise until the solution pH was increased to 10. The mixture was then stirred for a further 2 h. The product was collected by centrifugation and dried in an oven for 6 h at 120 °C. The white powder was then calcined at 500 °C in a furnace.

### 2.3. Processing of Hybrid Composites

The biopolymers were dried in an oven at 80 °C overnight before processing. Preliminary optimization of the blends was carried out at different ratios, i.e., PBS/PBAT (90/10, 80/20, 70/30, 60/40, and 50/50). The optimum ratio of PBS to PBAT blend that was selected was 70:30. The lignosulphonate loadings ranged from 5 to 15 wt.%, whereas the content of ZnO nanoparticles was kept constant at 1.5 wt.%. The designations and compositions of the samples are shown in Table 2.

All of the samples were extruded using a co-rotating twin-screw extruder with L/D 40:1, model: TE-30/600-11-40. The heating zone temperatures of the extruder ranged from 120 to 145 °C. The extruded samples were pelletized and dried in an oven at 80 °C for 24 h. The dried pellets of the samples were transferred to an injection moulding equipment (ENGEL e-mac50, ENGEL AUSTRIA GmbH, Schwertberg, Austria) with a 500-kN to produce dog bone test specimens, as depicted in Figure 1. The heating zone temperatures of the injection moulding ranged between 120 and 145 °C. The injection moulded dog bone specimens were cooled at room temperature and stored in a zip lock bag for further analysis.

### 2.4. Self Cleaning Measurements

The test solution was prepared by dissolving 15 mg of methylene blue into 100 mL of distilled water. For the individual experiments, an aliquot of 10 mL was put into a 10 mL glass beaker along with 100 mg of the polymer blend composite films, which were cut into ca. 15 mm × 5 mm strips. The lamp (165 W, with a radiation with λ less than 400 nm and an intensity of 240 W/m^2^) was placed such that the end of the lamp was 20 cm above the reactor vessel. The reaction solution was stirred for 2 h and an aliquot of the solution was drawn using a syringe and was analysed on a HPLC-PDA, as described below.

#### High Performance Liquid Chromatography (HPLC)

The residual concentration of the methylene blue after the self-cleaning tests were conducted versus the original concentration of the methylene blue was considered. The method used was adopted from a report available in the literature, as cited here [33]. In this instance, the methylene blue sample before and after the photocatalytic experiments was 10 µL in a Phenomenex LUNA 5 μm C18 reverse phase column (150 × 4.60 mm). The methylene blue was eluted using an isocratic method made from 45% HPLC grade acetonitrile and 45% Milli-Q water spiked with 0.1% *v*/*v* formic acid. The methylene blue was eluted at 0.02 mL/min. The response on the PDA detector was measured at 665 nm. The instrument used was a Shimadzu HPLC, SPD-M20A (Shimadzu, Roodepoort, South Africa), and the prominence was measured with a UV/VIS Photodiode Array Detector.

### 2.5. Antimicrobial Tests

The antimicrobial activity of the polymer hybrid composites was evaluated using the dynamic shake flask test method (American Society for Testing and Materials (ASTM) E2149-10) [34], with minor modifications. The working bacterial suspension was prepared by matching the turbidity of *E. coli* (approx. 1.5 × 10^8^ CFU/mL) suspended in deionized water to a 0.5 Mcfarland standard. This suspension was further diluted to approx. 1.5 × 10^7^ CFU/mL by placing 500 µL in sterilized test tubes containing 4.5 mL of deionized water. Strips (2.5 cm × 2.5 cm) of the polymer hybrid composites were inserted into the test tubes. After vertexing for 3 s, the test tubes were incubated with continuous shaking at 37 °C for 48 h. After incubation, 20 µL was placed on nutrient agar to determine the surviving *E. coli* cells. *E. coli* suspended in deionized water was used as a growth control, while *E. coli* suspended in the presence of a neat polymer blend strip was used as a negative control. The experiments were performed in triplicate.

### 2.6. Characterization Techniques

#### 2.6.1. Fourier Transform Infrared (FTIR) Spectroscopy

The FTIR spectra of the lignin, biopolymer blends, and hybrid composites were obtained using an ATR 4000 spectrophotometer (PerkinElmer, Waltham, Massachusetts, United States of America). All of the samples were scanned over a range of 4000–600 cm^−1^ with an average of 16 scans with a resolution of 4 cm^−1^.

#### 2.6.2. Tensile Properties

Five specimens from each biopolymer (PBS and PBAT), biopolymer blends (PBS/PBAT), and hybrid composites reinforced with various loadings (5, 10, and 15 wt.%) of lignin and 1.5 wt.% ZnO nanoparticles were tested for mechanical properties after they were conditioned at room temperature (25 °C) for 48 h. The tensile test was carried out using an Instron tensile tester, in compliance with the standard ASTM D638, at room temperature (25 °C).

#### 2.6.3. Thermal Gravimetric Analysis (TGA)

Thermogravimetric analysis was carried out on a thermogravimetric analyser (PerkinElmer, Buckinghamshire, UK) at a heating rate of 10 °C/min using nitrogen as a purge gas to analyse the lignin, biopolymer blends, and composites. The TGA was conditioned to increase the temperature linearly from room temperature to 900 °C under nitrogen flow. The temperature of the sample was monitored, and the sample’s weight loss was expressed in terms of percentage weight loss.

#### 2.6.4. Differential Scanning Calorimetry (DSC)

The differential scanning calorimetry (DSC) measurements on the biopolymer blends and composites were performed with a (DSC-Q2000 (TA Instruments, New Castle, DE, USA)) to evaluate the thermal transitions. In this regard, the cold crystallization and melting temperatures together with their corresponding enthalpies were investigated. DSC heating and cooling measurements of the samples were performed for two cycles at a flow rate of 25 mL/min under nitrogen gas. The samples were heated from −60 to 200 °C at a heating rate of 10 °C/min, cooled to −60 °C, and then heated to 200 °C. The analysis for the heating cycle was carried out using the second heating scan, whereas the cooling scan was used to evaluate the crystallization of the samples from the melt.

#### 2.6.5. X-ray Diffraction (XRD)

The XRD profiles of the blend and hybrid composites were measured using an Anton Paar SAXSess (Anton Paar, Graz, Austria) instrument operating at 40 kV and 50 mA in line collimation. The radiation was Cu Kα radiation with a wavelength of 0.154 nm and the exposure time of the sample was 10 min. Scattering radiation was detected in a 2θ = 5–90° rate of 1 s per step.

#### 2.6.6. Transmission Electron Microscopy (TEM)

The dimensions of the ZnO nanoparticles were determined using transmission electron microscopy (TEM, FEI Tecnai G2 Spirit electron microscope at 120 kV, FEI company, Hillsboro, OR, USA).

## 3. Results and Discussions

### 3.1. Chemical Characterization

Figure 2 shows the FTIR spectra for the lignosulphonate, blend, and hybrid composites reinforced with various lignin loadings (5 to 15 wt.%) and ZnO nanoparticles (1.5 wt.%). With regards to lignin, it is a complex polymer with many functional groups, such as sulphonic acid, methoxyl, phenolic, carboxyl, ketones, and carbonyl groups [35]. The strong absorption band at 3388 cm^−1^ assigned to OH stretching and the absorption band at wavenumbers of 2974 cm^−1^ corresponds to the methyl and methylene groups of lignin. The strong absorption band at 1571 cm^−1^ and 1417 cm^−1^ corresponds to the aromatic structure of lignin [35,36]. The absorption bands at 1185 cm^−1^ and 1117 cm^−1^ belong to the ester bonds of p-hydroxyphenyl propane units and ether–*O* for the syringyl structures, respectively, while the band at 1020 cm^−1^ corresponds to the –CH=CH– group out of plane deformation [36,37]. The bands between 600 cm^−1^ and 700 cm^−1^ represent the sulphonic groups [35]. In the case of a blend, the bands at 2945 cm^−1^ and 1708 cm^−1^ correspond to asymmetric stretching of the CH groups and carbonyl, respectively. The band at 1165 cm^−1^ corresponds to –C–O–C– stretching of the ester bonds [17]. With regards to the hybrid composites, their spectra displayed the same main bands presented in the blend. However, the hybrid composites demonstrated some differences in their spectra in comparison with the spectrum of a blend. For instance, all of the composites showed a band at around 3400 cm^−1^. This band was not clearly seen in the blend, but it was visible in the lignin. The presence of these bands could be attributed to the OH stretching. In addition, the band at 1584 cm^−1^ was pronounced in all of the composites. A similar absorption band could be observed in the case of lignin. However, the same band was not evident in the blend. This band could be attributed to the aromatic structure of lignin. This suggests that there was a non-covalent bond between the lignin and the blend. Similar findings were reported before for a ternary system comprised of PBAT, PLA, and lignin [38]. In their study, they noticed that the aromatic skeletal was shifting in higher wavenumbers, which confirms that there were some non-covalent interactions between the lignin and the blend. In fact, the shift indicated that the bonding between the lignin and the blend is more than just hydrogen bonding between the hydroxyl groups of lignin and carbonyl groups of the blend [38].

### 3.2. Mechanical Properties

Tensile tests were carried out on the dog bone specimens, and the results are listed in Table 3. From the results, it was observed that PBS was stiffer with a high tensile modulus and low elongation at break, while PBAT was ductile with a low tensile modulus and very high elongation at break. As anticipated, the introduction of PBAT into PBS decreased the tensile strength and modulus of the resultant blend, as well as elongation at break. To increase the PBAT content further resulted in further decreased tensile strength and modulus, while the elongation at break was increasing (Appendix A). A blend ratio of PBS and PBAT (70:30) was selected because it was comparable in terms of performance with the commercially available high-density polyethylene (HDPE). The tensile properties (tensile strength, tensile modulus, and elongation at break) of the blend and hybrid composites developed, represented in Table 3, displayed a similar behaviour to that which was reported in other studies when lignin was incorporated in neat PBS and in the blend of PBAT and poly(lactic acid) (PLA), respectively [5,38]. The decrease in tensile strength and elongation at break was noticeable in the hybrid composites. However, increasing the filler loading resulted in a further decrease in tensile strength and elongation at break. The decrease in both tensile strength and elongation at break was more pronounced when a great amount of fillers was introduced. The tensile strength and elongation at break when 15 wt.% was added in the blend were 22.57 MPa and 32.94%, which were decreased by approximately 18 and 167%, respectively. This drop in properties was attributed to the low aspect ratio of the filler, which is comprised of aromatic rings [5]. In contrast, the stiffness of the resultant hybrid composites represented by the tensile modulus increased. Similar results were reported in another study [27]. This stiffness was mainly due to the introduction of rigid filler particles. In addition, when the filler loading increased further, the tensile modulus of the resultant hybrid composites gradually increased. The tensile modulus when 15 wt.% was added was 347.44 MPa, which improved by approximately 31%, which was close to that obtained from the neat PBS.

### 3.3. Thermal Properties

In Figure 3a,b, the results of the TGA and derivative curves of the lignin, blend, and hybrid composites are reported. The thermal characterizations of these materials are summarized in Table 4. ZnO nanoparticles remained stable at baseline within 900 °C (Appendix A), whereas the lignin, blend, and hybrid composites degraded when the temperatures increased to 900 °C. The thermal degradation of lignin occurred in three steps. The lignin thermogram showed a small weight loss in the region of 50–120 °C, which is due to the evaporation of moisture and the low molecular weight materials [36]. During the heating of lignin in the temperature range between 120 °C to 900 °C, a huge mass loss was observed at 265.7 °C and 729.1 °C, which was attributed to the release of volatiles such as carbon dioxide and methane, in addition to SO_2_ and mercaptans from the lignin [35]. In addition, the main degradation temperature of lignin shifted to lower temperatures in comparison with the blend and hybrid composites, with high char residues after degradation. This indicates that lignin is thermally less stable when compared with the aforementioned materials. The char residue in the case of lignin was attributed to the complex aromatic nature of the lignin [36]. In the case of the blend and hybrid composites, the degradation occurred in a single major weight loss step. It was noticeable that the blend was thermally more stable in comparison with the hybrid composites. The incorporation of a thermally less stable lignin led to a decrease in the thermal stability of the hybrid composites. A previous study [39] also reported comparable results. In that study, they incorporated alkaline soda lignin into PBAT and investigated the thermal stabilities. The results revealed that the addition of lignin shifted the maximum temperatures of PBAT to lower temperatures. Furthermore, hybrid composites had a higher char residue in comparison with the blend, which was attributed to the incorporation of lignin with an aromatic structure, which is able to form a stable char upon pyrolysis [39].

DSC curves related to the second heating and cooling of biopolymer blends and hybrid composites are shown in Figure 4a,b. The glass transition temperature (T_g_) of the blend and hybrid composites was not visible. The incorporation of fillers into the blend did not change the melting behaviour of the materials, as shown in Figure 4a and Table 4. Their melting temperatures (T_m_) remained the same (115 °C). However, when increasing the lignin loading from 5 wt.% to 15 wt.%, T_m_ did not show any alterations. A similar trend was reported in other studies [5,18], where lignin was introduced into PBS and PBAT, respectively. The melting and crystallization enthalpies, as well as cold crystallization temperature (T_cc_), decreased when the filler was incorporated. In addition, when the filler loading increased, the melting and crystallization enthalpies and T_cc_ further decreased. Sahoo et al. [40] reported comparable results. In their study, they investigated the effect of lignin on the crystallization and melting behaviour of PBS composites reinforced with lignin. The incorporation of lignin increased T_g_, which suggests a good interaction between the polymer and filler. The addition of up to 50 wt.% increased the crystallinities of the material. Moreover, the addition of lignin, irrespective of loading, had no effect on the melting behaviour of PBS.

### 3.4. XRD and TEM Analysis

The morphology of the synthesized ZnO particles was found to be largely hexagonal, with some slightly spherical particles present, as shown in Figure 5a. The diameter of the ZnO nanoparticles was estimated by measuring at least 100 particles using ImageJ, and it was found to range between 60 and 110 nm, with most of them having an average size of 80 nm. The PXRD pattern of the ZnO is shown in Figure 5b. The pattern is consisted with that of hexagonal wurtzite (JCPDS No. 36–1451). The reflections observed were at 2θ positions of 37.14, 40.07, 42.57, 56.10, 66.5, 74.66, 78.93, 80.97, and 82.322°. These were indexed as (100), (002), (101), (102), (110), (103), (200), (112), and (201), respectively. There were no other peaks observed, suggesting that the product was pure. The PXRD pattern of the blend was characterized by peaks at 2θ positions of 17.48, 19.70, 22.55, and 28.93. These peaks are characteristic of PBS, as was reported by Tang et al. [41], who reported a similar XRD pattern. The peaks associated with PBAT were not observed, as PBAT is much more amorphous when compared with PBS, and PBAT formed a lower ratio of the blend, 30 wt.%. The amount of ZnO nanoparticles added to the blend was 1.5 wt.%; although it is crystalline, the weight percentage was too low to be observable on the PXRD pattern. This meant that the PXRD of the blend was similar to that of PBS and the hybrid composites. This indicates that the addition of PBAT, lignin, and ZnO nanoparticles did not influence the crystal structure of the PBS matrix.

### 3.5. Self-Cleaning

Figure 6a,b shows the peak intensities and the removal efficiencies, respectively, of the photodegradation after the 2 h experiments. Here, it was observed that the intensity of the original methylene blue (not photodegraded) was similar to that of the methylene blue that was photodegraded using the blend. The lack of photocatalytic activity for the blend was attributed to the fact that the materials that make up the blend, i.e., PBAT and PBS, do not have photocatalytic properties. The blends with lignin and ZnO nanoparticles all performed in a comparable way, showing a removal efficiency greater than 70%. Figure 6c is a picture of the methylene blue test solution after the photodegradation experiments. The picture on the left is of the experiment conducted with blend + lignin 15 wt.%, while the picture on the right was conducted with just the blend. The picture on the left is almost clear, showing the removal of methylene blue, while the one on the right remained blue, showing that the blend was not photocatalytic active.

### 3.6. Antimicrobial Studies

To investigate if the polymer blends possessed antimicrobial properties, the shake flask method test method (ASTM E2149-10), a standard method used for determining the antimicrobial activity of immobilized agents, was used. *E. coli* (Gram-negative bacterium) was incubated in deionized water in the presence of the polymer blend strips. The growth control experiment was conducted without the addition of polymer strips. After incubation for 18–24 h, the surviving *E. coli* cells were determined by placing 20 µL on nutrient agar plates and being incubated for 48 h. The results of the antimicrobial activity experiments are depicted in Figure 7. The polymer hybrid composites displayed an excellent antimicrobial activity against *E. coli* (Figure 7c–e), as no growth of *E. coli* was observed. These results agree with the antimicrobial activity of the ZnO NP-loaded polymer blends reported by others [42,43,44,45]. However, the loading of the ZnO Np reported was higher (1–6 wt.%) than that reported herein (1.5 wt.%). Both the growth and the negative control experiments (Figure 7a,b) showed the growth of *E. coli*, an indication that the observed antimicrobial activity was imparted by the ZnO nanoparticles, and possibly the presence of lignin. Lignin has been reported to possess an antimicrobial activity against a broad spectrum of Gram-negative and Gram-positive bacteria [46]. This suggests that there could be a synergistic effect between lignin and the ZnO nanoparticles towards the inhibition of the growth of *E. coli*. This phenomenon can be investigated, in the future, to establish its existence or lack thereof.

## 4. Conclusions

In summary, lignin and ZnO nanoparticles were introduced into biopolymer blends to reduce their cost and improve their performance. PBAT was added to PBS in an attempt to reduce its stiffness. The lignin loading was optimized up to 15 wt.%. It is imperative to note that the processes (hot melt extrusion and injection moulding) used to fabricate hybrid composites are environmentally benign, as no toxic chemicals were used, and they are industrial scalable. In addition, both PBS and PBAT are biodegradable polymers; however, the introduction of lignin and ZnO nanoparticles does not compromise their biodegradability feature. Therefore, the combination of biopolymers and these fillers produce green materials or biocomposites. The introduction of fillers improved the stiffness of the composites. Additionally, the hybrid composite displayed a lower tensile strength and elongation at break when the fillers were introduced. In addition, the incorporation of fillers decreased the thermal stability of the resultant materials. Furthermore, the results showed that the resultant hybrid composites showed both self-cleaning and antimicrobial properties. The properties presented in this study have shown that these materials have potential use for biomedical applications. The research on lignin-based biocomposites for biomedical applications is still in an infancy stage. However, future research is required in order to realize the full potential of lignin-based biocomposites.

## Figures and Tables

**Figure 1 polymers-14-05065-f001:**
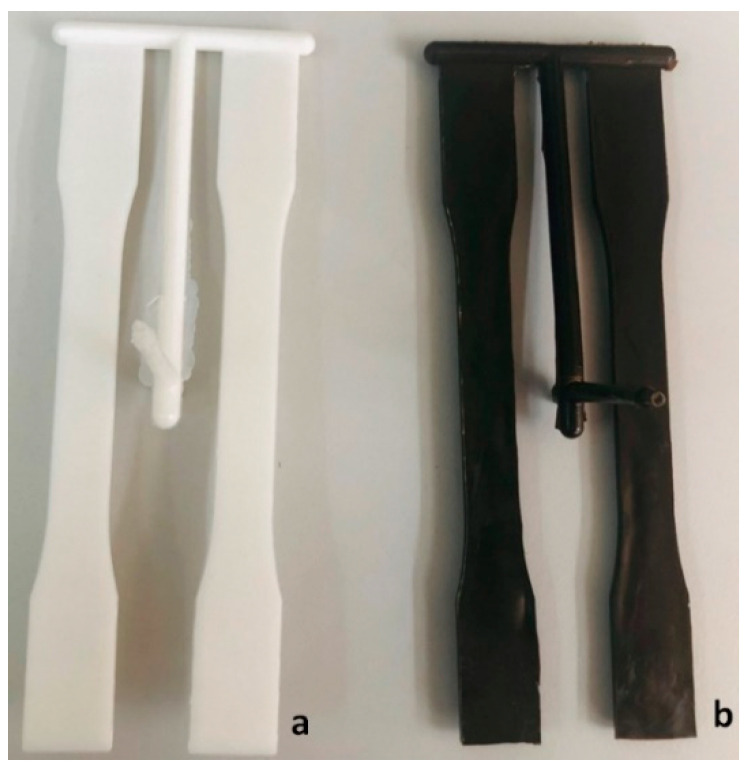
Image of dog bone test specimen prepared: (**a**) blend and (**b**) composites reinforced with 15 wt.% lignin.

**Figure 2 polymers-14-05065-f002:**
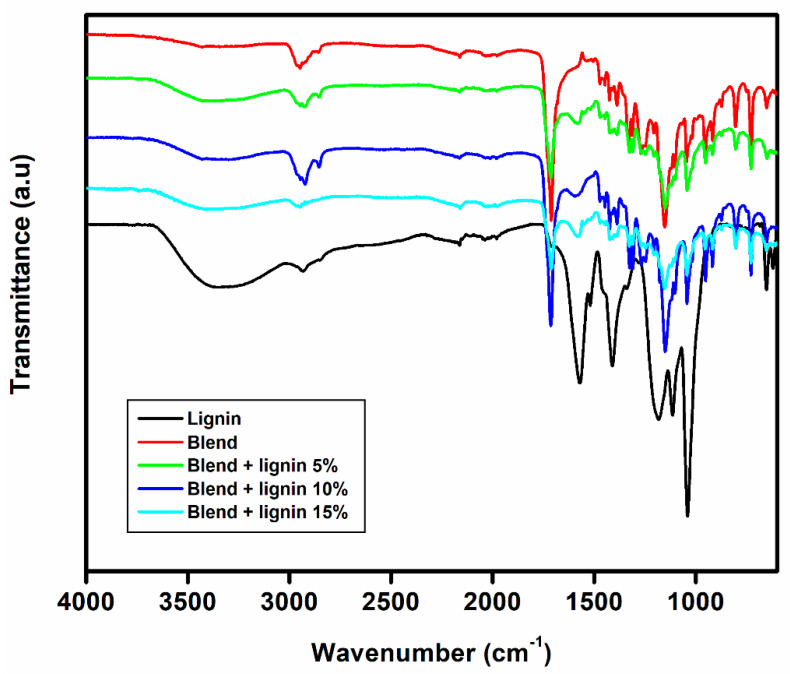
FTIR spectra of lignin, polymer blends, and composites.

**Figure 3 polymers-14-05065-f003:**
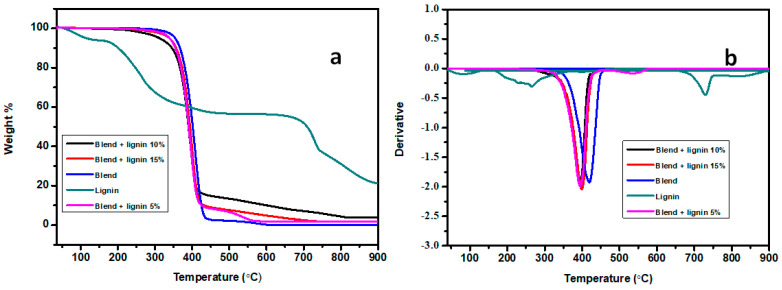
(**a**) TGA and (**b**) DTG curves of the lignin, biopolymer blend, and composites.

**Figure 4 polymers-14-05065-f004:**
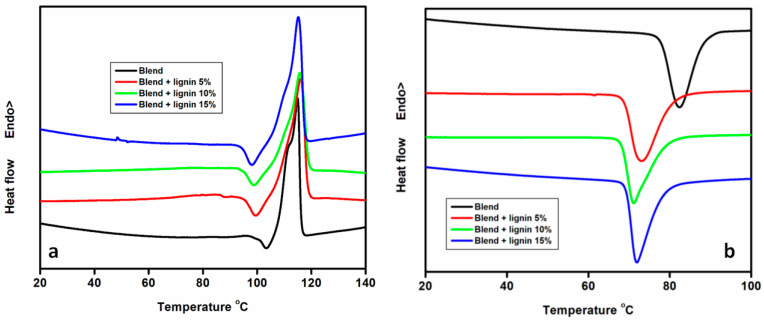
DSC thermograms of the blend and composites after (**a**) heating and (**b**) cooling.

**Figure 5 polymers-14-05065-f005:**
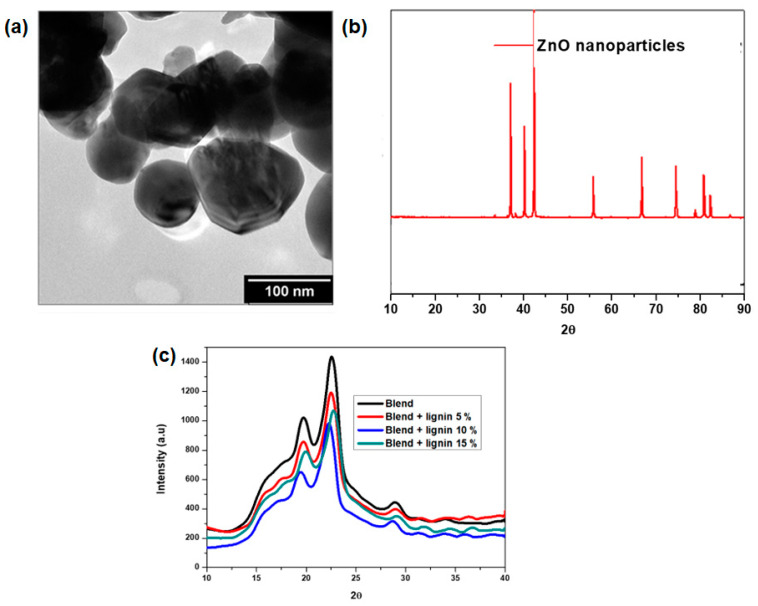
(**a**) TEM image of ZnO nanoparticles, (**b**) XRD profile of ZnO nanoparticles, and (**c**) XRD profiles of biopolymer blend and hybrid composites.

**Figure 6 polymers-14-05065-f006:**
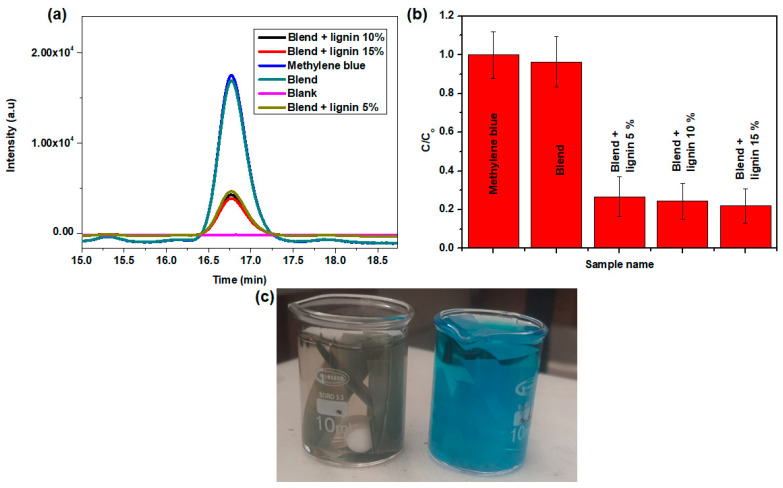
(**a**) Chromatogram of methylene blue, (**b**) photocatalytic efficiency of the various materials, and (**c**) methylene blue test solutions after 2 h photodegradation experiments using (**left**) blend + lignin 15% and (**right**) blend.

**Figure 7 polymers-14-05065-f007:**
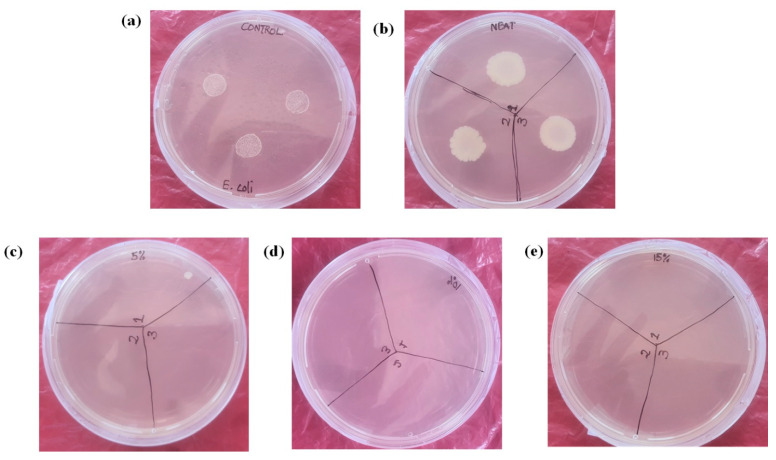
Images of petri dishes of the antimicrobial tests on various samples: (**a**) control, (**b**) blend, (**c**) blend reinforced with 5 wt.% lignin, (**d**) blend reinforced with 10 wt.% lignin, and (**e**) blend reinforced with 15 wt.% lignin.

**Table 1 polymers-14-05065-t001:** Biopolymers used in this study for the development of composites.

Polymer	Trade Name	Company and Country
Poly butylene adipate-co-terephthalate (PBAT)	Eco flex C1200	BASF, Ludwigshafen, Germany
Biobased polybutylene succinate (PBS)	Bio PBS FZ 91	PTT MCC BIOCHEM, Chatuchak, Bangkok, Thailand

**Table 2 polymers-14-05065-t002:** Different formulations of the samples.

(PBS/PBAT)/Lignin/ZnO	PBS/PBAT	Lignin	ZnO
100/0/0	100	0	0
95/5/1.5	95	5	1.5
90/10/1.5	90	10	1.5
85/15/1.5	85	15	1.5

**Table 3 polymers-14-05065-t003:** Tensile properties of hybrid composites with different lignin loadings.

Sample	Tensile Strength at Yield (MPa)	Tensile Strain at Break %	Tensile Modulus (MPa)
PBS	37.67 ± 0.83	457 ± 24.02	357 ± 15.47
PBAT	8.38 ± 0.11	1040 ± 9.76	52.01 ± 28.78
PBS-PBAT (70/30)	26.99 ± 0.32	367.01 ± 86.87	253.49 ± 13.40
(PBS-PBAT (70/30)/(lignin) (95/5))	26.05 ± 0.47	145.48 ± 67.54	261.88 ± 14.78
(PBS-PBAT (70/30)/(lignin) (90/10))	24.03 ± 0.26	138.62 ± 57.88	304.52 ± 5.22
(PBS-PBAT (70/30)/(lignin) (85/15))	22.57 ± 0.37	32.94 ± 5.88	347.44 ± 16.92

**Table 4 polymers-14-05065-t004:** Thermal characterization of the blends and their composites from TGA and DSC.

Sample	T_max_	T_m_	∆H_m_	T_cc_	∆H_c_
Lignin	82.5; 265.7; 729.1	-	-	-	-
PBS-PBAT (70/30)	409.2	114.9	50.6	82.4	56.9
(PBS-PBAT (70/30)/(lignin) (95/5))	394.5	115.7	46.9	73.1	49.9
(PBS-PBAT 70/30)/(lignin) (90/10))	394.7	115.7	43.5	71.2	45.2
(PBS-PBAT 70/30)/(lignin) (85/15))	398.6	115.2	45.4	71.9	44.1

## Data Availability

The data presented in this study are available on request from the corresponding author.

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
