# Peer review of "Fabrication of a Polybutylene Succinate (PBS)/Polybutylene Adipate-Co-Terephthalate (PBAT)-Based Hybrid System Reinforced with Lignin and Zinc Nanoparticles for Potential Biomedical Applications"

_polymers, 2022, doi:10.3390/polym14235065_

Round 1
Reviewer 1 Report
In this work the authors prepared PBAT/PBS, the thermal stability of which can be improved by blending with lignin. Furthermore, the authors investigated the composite of PBAT/PBS@ZnO, which possesses photocatalytic performance, leading to serving as self-cleaning materials. The work is done well, so the referee supports its publication after minor revision as below.
1 The interaction between biocopolymer and lignin should be given.
2 The stability of the composite containing ZnO should be studied. Could it be recycle used for catalyst.
3 In TGA test (Figure 3a), why pristine lignin maintained the largest percentage of weight?
Author Response
Dear reviewer
Please find attached document.
Kind regards
Dr A Mtibe

Reviewer 2 Report
This manuscript reported a novel study upon preparing an composite of PBS and PBAT. Besides, the lignin and ZnO were used to reinforce the composite. The experiments carried out are well-designed, and the obtained results were detailed discussed, showing well organized manuscript. Taking into account the results obtained, the study represents promising environmentally friendly technology. The topic of this paper is interesting and the results could broad the application field of lignin. However, there are some minor issues (listed below) and should be revised before acceptance. Thus, in my opinion, this manuscript could be accepted after minor revision.
Specific comments
1. It is suggested to add more detailed introduction of previous research about lignin and ZnO for reinforcing composite, and described the problems which need to be solved.
2. Please add the illustrations for figure a and b and others in the titles of figure 1,3,4,7.
3. It is suggested to combine the part of 2.4 and 2.5.
4. Data not shown in line 232 and 256 could be presented in supplemental material.
5. According to the DTG result, lignin has been thermally decomposed below 200 oC, while the samples include lignin were heated from -60 to 200 oC two time during DSC measurement. So, is it possible that the first heating in DSC would change the chemical structure and affect the result in second heating?
6. If possible, please add the DSC measurement of lignin in figure 4.
Author Response
Dear Reviewer,
Please see attached document.
Kind regards
Dr A Mtibe
